# EXERCAKT: A KNOWLEDGE TRACING MODEL BASED ON GRU CAPTURING CONTEXTUAL FEATURES OF EXERCISES

## ABSTRACT

Knowledge tracing aims to predict students' future performance based on their past interactions, helping online learning platforms and teachers assess learners' knowledge levels. This technology plays a critical role in achieving large-scale cognitive diagnosis. Recently, deep learning-based knowledge tracing models have demonstrated impressive results, with most research focusing on designing customized network architectures and novel optimization objectives. However, redundant parameters and overly complex loss functions can complicate model training and make it harder to maintain prediction accuracy.To further investigate the effectiveness of simple recurrent neural networks in this field, and to leverage their advantages in handling sequential exercise representation, this paper introduces a GRU-based knowledge tracing model named ExerCAKT (Exercise Context-Aware Knowledge Tracing). This model effectively captures contextual features of exercises and achieves robust knowledge state modeling through the use of a GRU-based knowledge state feature extractor and a GRU-based exercise feature extractor—without relying on additional optimization objectives.The model's superior performance is validated through comparisons with baseline models, such as AKT and SIMPLEKT, on three public datasets in the knowledge tracing domain. Evaluations are conducted using AUC and ACC metrics at both the Knowledge Concept level and the question level. We validated that relying solely on simple recurrent neural networks, combined with appropriate representation methods, can still achieve excellent performance in this field. Our code will be available at xxx (Anonymous URL).

## 1 INTRODUCTION

Knowledge tracing is a foundational technology for achieving large-scale personalized education, helping educators diagnose learners' cognitive levels to provide more targeted teaching. In recent years, knowledge tracing has seen widespread application in online learning platforms (Abdelrahman et al., 2023; Song et al., 2022a; Käser et al., 2017). The task of knowledge tracing is illustrated in Figure 1. Students complete exercises in an online learning system, and each student's responses to these exercises are referred to as interactions in knowledge tracing. Each interaction generates at least the following data: exercise ID, knowledge concept ID, and whether the student answered the question correctly. All exercises are associated with knowledge concepts, and an exercise may be linked to multiple knowledge concepts, as shown with Q4, Q5, and Q8 in the figure. To predict a student's performance in the next interaction, the input to the knowledge tracing algorithm typically includes the student's historical interaction information and the information of the exercise to be predicted. Therefore, better modeling of interaction information and representation of the exercises to be predicted are crucial strategies for achieving good performance.

Recently, Liu et al. (2022) introduced a tool for standardized experiments in the knowledge tracing field called pyKT, which implemented multiple advanced knowledge tracing models under various experimental conditions and published the optimal results. The benchmark provided by pyKT holds significant reference value. We meticulously reviewed this work and the publicly available code, and replicated their experiments. Based on these results, we obtained the same and surprising conclusion: the first model to apply deep learning to the knowledge tracing field, DKT (Piech et al.,

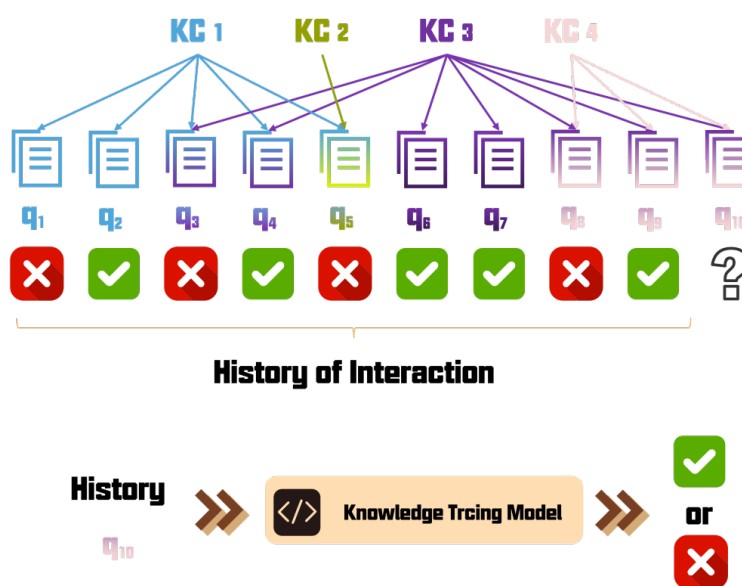

Figure 1: The form of the knowledge tracing task and the model's input and output.

2015), still achieves excellent performance under standard experimental conditions.We are excited to see DKT achieve outstanding results, as it is an extremely simple model, suggesting that we may not need overly complex models to achieve excellent performance in knowledge tracing tasks.

The success of ChatGPT has generated significant attention (Wu et al., 2023), and its underlying Transformer (Vaswani et al., 2017) had already demonstrated excellent performance in the field of natural language processing with the advent of language models like BERT (Devlin et al., 2019) and T5 (Raffel et al., 2020). Subsequently, Transformer models have been applied in computer vision and recommendation systems, with representative works such as ViT (Dosovitskiy et al., 2020), which has cross-modal modeling capabilities for multimodal learning, and SASRec (Kang & McAuley, 2018), a Transformer-based model that effectively improves the accuracy of recommendation algorithms.

However, Transformer models have not achieved breakthrough success in the field of knowledge tracing. The representative method, SAINT (Choi et al., 2020), uses a Transformer-based encoder and decoder to capture features of student interaction sequences, but the improvements have been limited. In the comparative experiments of pyKT (Liu et al., 2022), evaluations on seven different datasets under the KC Level evaluation mode showed poor performance in more than half of the datasets. This somewhat supports our view that in the field of knowledge tracing, the complexity of model structures does not necessarily lead to better results. By analyzing the similarities and differences between the field of knowledge tracing and other fields such as natural language processing, computer vision, and recommendation systems, we believe that the reasons Transformer-based knowledge tracing models fail to achieve good results may include: (1) The knowledge tracing field is more sensitive to sequence positions, and the "token embedding + positional encoding" method disrupts the underlying semantic information of interaction encoding and reduces the benefits brought by sequence information. (2) The classic knowledge tracing datasets generally have fewer interactions, and too many parameters may lead to overfitting issues. Therefore, we believe that using recurrent neural models, which can naturally capture temporal relationships, along with appropriate knowledge state and exercise representations, are two ways to achieve better results in the field of knowledge tracing.

In this paper, we move away from the current obsession with "Attention is all you need" and design our model ExerCAKT based on gated recurrent networks (Chung et al., 2014). The model includes an embedding layer, a knowledge state feature extractor, an exercise feature extractor, and a prediction layer used to predict the student's performance in the next exercise. The contributions of this paper are as follows: (1)Design of a GRU-based Exercise Feature Extractor: We have designed

a GRU-based exercise feature extractor and experimentally validated its effectiveness and superior performance. (2)Proposal of a Recurrent Neural Network-Based Knowledge Tracing Model: We propose a knowledge tracing model based on a recurrent neural network architecture. By utilizing the designed GRU feature extractor, the model captures contextual features of exercises, achieving modeling of knowledge states and high-quality representation of exercises. ExerCAKT explores the potential of a simple recurrent neural network architecture in knowledge tracing tasks and maximizes the advantages of recurrent neural networks in representing sequential data. (3)Significant Improvement Over Baseline Models: Compared to baseline models, ExerCAKT shows substantial improvements. Experiments on the ASSISTments2009, Algebra2005, and ASSISTments2015 datasets at both the KC-level and Question-Level outperform strong baseline models like AKT (Ghosh et al., 2020) and SIMPLEKT (Liu et al., 2023b). The results demonstrate that our model has competitive predictive performance compared to attention-based methods. We confirm that methods based on recurrent neural network architectures still have significant potential and exploration space in the field of knowledge tracing.

## 2 RELATED WORK

### 2.1 PROBLEM DEFINITION

The goal of knowledge tracing is to predict a student's performance in future interactions based on their interaction sequence, which can be viewed as a sequence modeling task. In this task, the student's learning history is considered as an interaction sequence, typically representing the history of the student's responses to exercises. The model's objective is to predict whether the student will answer the next question correctly based on the data in this sequence. Formally, let the interaction sequence be denoted as $S$, and the interactions that have occurred up to time $t$ be represented as $S = [R_0, R_1, \ldots, R_t] = [(q_0, c_0, r_0), (q_1, c_1, r_1), \ldots, (q_t, c_t, r_t)]$, where each interaction records the exercise ID $q_t$, the concept $c_t$, and whether the response was correct $a_t$ (i.e., 0 or 1). The goal of knowledge tracing is to predict the student's response $r_{t+1}$ given the exercise $e_{t+1}$ and concept $c_{t+1}$ at the next time step.

### 2.2 DEEP LEARNING-BASED KNOWLEDGE TRACING

Currently, deep learning has been widely applied in the field of knowledge tracing. The work in this area can be roughly categorized into methods based on recurrent neural networks, memory networks, self-attention mechanisms, and other neural network methods. Subsequent work based on these four categories primarily focuses on improving methods through three aspects: optimizing the network architecture, adding additional features, and simulating the real learning processes of humans.

#### 2.2.1 METHODS BASED ON RECURRENT NEURAL NETWORKS

DKT (Piech et al., 2015) is the first model to use deep learning techniques in the field of knowledge tracing. It directly employs Long Short-Term Memory (LSTM) networks to model students' interaction sequences and predict their performance in subsequent interactions. Subsequent work such as DKT+ (Yeung & Yeung, 2018) and KQN (Lee & Yeung, 2019) builds on DKT by improving the training objectives and network architecture for methods based on recurrent neural networks.

#### 2.2.2 METHODS BASED ON MEMORY NETWORKS

Methods based on memory networks still use a recurrent structure. Compared to LSTM, memory networks use matrices to store historical information, which provides better long-term memory capabilities. A representative work in this category is the Dynamic Key-Value Memory Network (DKVMN) (Zhang et al., 2017).

#### 2.2.3 METHODS BASED ON SELF-ATTENTION MECHANISMS

Self-attention mechanisms involve calculating the importance of each input element relative to other elements, thereby assigning different weights to each element. This allows the model to dynamically adjust attention based on the internal relationships of the data when processing sequence

data, thereby focusing on information from different positions with strong expressive capability. SAKT (Pandey & Karypis, 2019) is the first knowledge tracing framework based on self-attention mechanisms. Other notable works in this area include SAINT (Choi et al., 2020) and AKT (Ghosh et al., 2020).

### 2.2.4 METHODS BASED ON OTHER NEURAL NETWORKS

In addition to the three basic methods mentioned above, some works have introduced other network structures or training methods. For example, GKT (Nakagawa et al., 2019)and Bi-CLKT (Song et al., 2022b) use graph neural networks to model the dependencies between knowledge concepts, ATKT (Guo et al., 2021) employs adversarial learning to enhance generalization capabilities, and CL4KT (Lee et al., 2022) utilizes contrastive learning. Currently, these four types of methods have established stable research communities, with relatively mature research directions. Introducing other networks or training methods is an important avenue for future exploration in the field of knowledge tracing.

## 3 MODEL

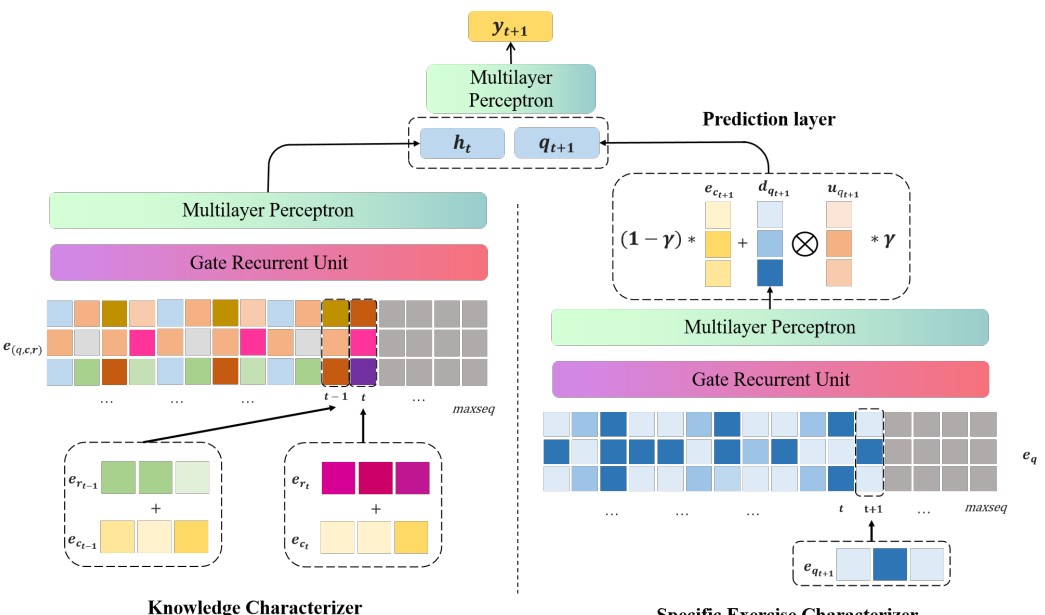

Figure 2: ExerCAKT model.

Our proposed method is illustrated in Figure 2 and includes the Embedding layer, Knowledge State Feature Extractor(Knowledge Characterizer), Exercise Feature Extractor(Specific Exercise Characterizer), and Prediction layer. The Knowledge State Feature Extractor is used to represent the learner's knowledge state, while the Exercise Feature Extractor extracts contextual information of the exercises to reflect the specificity of the exercises in different interaction sequences.

### 3.1 EMBEDDING LAYER

Similar to other KT models, each interaction of the learner includes the following information: (1)$q_t$, the exercise answered by the learner at time t; (2) $r_t$, whether the learner answered correctly; (3) $c_t$, the knowledge concept corresponding to the answered exercise. First, embeddings are created for all knowledge concepts and exercises, resulting in embedding matrices $E_c \in \mathbb{R}^{C \times d}$ and $E_q \in \mathbb{R}^{Q \times d}$, where $C$ and $Q$ denote the number of knowledge concepts and exercises, respectively, and $d$ represents the embedding dimension, which is consistent across all embedding matrices. For example, at time $t$, the representations of $q_t$ and $c_t$ are $e_{q_t}$ and $e_{c_t}$. Next, the learner's response

to interactions is embedded; since the response can only be "correct" or "incorrect", $E_r$ is set to $E_r \in \mathbb{R}^{2 \times d}$. Additionally, the parameter matrix $E_u \in \mathbb{R}^{Q \times 1}$ is used to represent the specificity sensitivity of the exercises, with details on specificity provided in subsequent section.

## 3.2 KNOWLEDGE STATE FEATURE EXTRACTOR

The purpose of the knowledge state feature extractor is to determine the student's knowledge mastery at different time steps based on the student's interaction sequence. To represent the learner's learning process, the interaction at time $t$ is denoted as:

$$e_{(q_t, c_t r_t)} = e_{c_t} + e_{r_t} \in \mathbb{R}^d \tag{1}$$

This means that modeling the learner's knowledge state is independent of the exercises $q_t$ answered at a given time step $t$. From a cognitive perspective, the reason for not using the exercise is that a learner's knowledge is primarily constructed around knowledge concepts. During the learning process, the learner mainly acquires and understands knowledge concepts rather than memorizing specific exercises. From a data perspective, the number of exercises is typically greater than the number of knowledge concepts, i.e., $Q > C$. The same knowledge concept often appears multiple times in the sequence, while an exercise typically does not reappear after being completed by the student. Therefore, the sparsity of sequence $[q_0, q_1, \ldots, q_t]$ is much higher than that of sequence $[c_0, c_1, \ldots, c_t]$. Incorporating $e_{q_t}$ into $e_{(q_t, c_t, r_t)}$ could increase the learning difficulty due to high sparsity and affect the representation effectiveness.

ExerCAKT uses standard Gated Recurrent Units (GRU) (Chung et al., 2014) as the sequence feature extractor. The reason for this choice is that the structure of GRU is similar to human memory. After each interaction occurs, GRU updates the knowledge state and reorganizes the knowledge structure. Additionally, GRU is simpler than LSTM. it combines the forget gate and input gate in LSTM into a single "update gate." This makes GRU have fewer parameters than LSTM, which generally results in faster training and inference. When dealing with shorter sequences, fewer parameters mean the network can learn important features more quickly, potentially leading to better performance. The formal description of GRU is as follows:

$$Z_t = \sigma \left( W_Z \cdot [h_{t-1}, x_t] \right) \tag{2}$$

$$r_t = \sigma \left( W_r \cdot [h_{t-1}, x_t] \right) \tag{3}$$

$$\tilde{h}_t = \tanh \left( W \cdot [r_t * h_{t-1}, x_t] \right) \tag{4}$$

$$h_t = (1 - Z_t) * h_{t-1} + Z_t * \tilde{h}_t \tag{5}$$

In GRU, $Z_t$ and $r_t$ are the update gate and reset gate weights, respectively, used for the update and reset operations. ExerCAKT uses $e_{(q_t, c_t, r_t)}$ as the input $x_t$ for GRU and then applies a multi-layer perceptron for feature transformation to enhance the model's non-linear processing capability, that is:

$$h_t = W_1 \left( \sigma \left( W_2 \, \text{GRU} \left( [\ldots, e_{(q_t, c_t, r_t)}] \right) \right) \right) \in \mathbb{R}^d \tag{6}$$

In this context, $W_1$ and $W_2$ represent neural network parameters, and $\sigma$ represent the GELU non-linearity (Hendrycks & Gimpel, 2016). Ultimately, the output $h_t$ at each time step contains information from the sequence over time $[0, t]$, which is used to represent the knowledge state features.

## 3.3 EXERCISE FEATURE EXTRACTOR

Although exercises are not used to describe the learning interaction sequence, learners interact based on the exercises they answer, so the representation of exercises is also important. The exercise feature extractor is used to generate the representation for the exercise at time $t + 1$. ExerCAKT does not directly use $e_{q_{t+1}}$ as the representation for the exercise at $t + 1$ for the following reasons:(1)When modeling knowledge states, only knowledge concept $c_t$ and answer correctness $r_t$ are used without specific exercises. This inconsistency might affect the model's performance.(2)As mentioned earlier, $Q > C$, and the high sparsity and long-tail distribution of exercises will lead to poor model performance when predicting exercises with lower frequency.(3)Directly using $e_{q_{t+1}}$

as the question representation does not consider the contextual relationships of exercises, thus neglecting non-knowledge factors such as related exercises and changes in difficulty when students are answering.

Through a review of the literature (Wright, 1977), the Rasch method treats exercises as "specific manifestations of knowledge concepts." Therefore, some studies (Ghosh et al., 2020; Liu et al., 2023b) represent exercises as a product of the exercise difficulty scalar $u_{q_t}$ and the knowledge concept difficulty vector $d_{c_t}$, plus the knowledge concept embedding $e_{c_t}$, i.e., $e_{q_t} = e_{c_t} + d_{c_t} * u_{q_t}$. In contrast to the aforementioned approaches, we use a combination of another GRU unit and a multi-layer perceptron to extract features from the exercise sequence $[q_0, q_1, \ldots, q_t, q_{t+1}]$. This feature is used as the specificity vector of the exercise concerning the knowledge concept (this vector represents the offset of the exercise with respect to the knowledge concept, indicating the exercise's specificity). This approach captures the contextual relationships of exercises to reflect the specificity of the exercises. Specifically,

$$d_{q_{t+1}} = W_3 \left( \sigma \left( W_4 \, \text{GRU} \left( [\ldots, e_{q_{t+1}}] \right) \right) \right) \in \mathbb{R}^d \tag{7}$$

At the same time, we set hyperparameters $\gamma$ to control the proportion between the specificity vector and the knowledge concept embedding vector, i.e.,

$$e_{q_{t+1}} = (1 - \gamma) e_{c_{t+1}} + \gamma * d_{q_{t+1}} * u_{q_{t+1}} \tag{8}$$

where $u_{q_t} \in \mathbb{R}^{Q \times 1}$ represents the sensitivity of the exercise to specificity; a larger value indicates greater consideration of the contextual relationship. Additionally, $u_{q_t}$ is initialized as a zero matrix to ensure high-quality representation of low-frequency exercises, thereby improving the model's robustness.

### 3.4 PREDICTION AND TRAINING

The prediction layer concatenates the knowledge state with the next time step's exercise to be predicted and uses a multi-layer perceptron and sigmoid function for the result prediction, i.e.,

$$\hat{y}_{t+1} = \sigma \left( W_5 \left[ h_t \oplus e_{q_{t+1}} \right] \right) \tag{9}$$

where $\oplus$ represents the vector concatenation operation.

ExerCAKT uses predicting the performance of the next time step interaction as the sole training objective, without introducing additional training objectives or regularization terms. It employs binary cross-entropy loss as the loss function, specifically:

$$\text{Loss} = -\sum_{t=1}^{T} \left( y_t \cdot \log \hat{y}_t + (1 - y_t) \log (1 - \hat{y}_t) \right) \tag{10}$$

where $y_t$ represents the true label of the student's interaction, and the Adam optimizer is used for optimization. The training objective is to minimize the loss.

## 4 EXPERIMENT

This paper first introduces the experimental setting, datasets, evaluation metrics, and baseline models used for comparison. In the main body, we report comparative analysis, ablation analysis and key hyperparameter analysis. Discussions on other hyperparameters and computing performance are included in the appendix, as are more details of the baseline model and datasets.

### 4.1 EXPERIMENTAL SETTING

The paper uses PyTorch version 1.10 and conducts experiments on four NVIDIA A100 Tensor Core GPUs. The general hyperparameters are selected as follows: batch sizes are $\{64, 128\}$, learning rates are $\{1e-3, 1e-4\}$, embedding dimensions are $\{64, 128, 256\}$, and dropout are $\{0.1, 0.2, 0.3, 0.4, 0.5\}$. Data preprocessing follows the standard procedure provided by pyKT (Liu et al., 2022): first, sequences with missing data or interaction lengths less than 3 are filtered out; then,

Table 1: datasets information

| Datasets | interactions | sequences | questions | KCs | avg KCs |
|---|---|---|---|---|---|
| ASSISTments2009 | 337,415 | 4,661 | 17,737 | 123 | 1.1970 |
| Algebra2005 | 884,098 | 4,712 | 173,113 | 112 | 1.3634 |
| ASSISTments2015 | 682,789 | 19,292 | - | 100 | - |

interactions are truncated to a maximum sequence length of 200; finally, interactions are evenly divided into 5 parts, with 4 parts used as the training set and 1 part used as the test set. In line with the pyKT experiment, we examine the AUC and ACC metrics for both KC-level and Question-Level (detailed explanations of KC-level and Question-Level can be found in reference (Liu et al., 2022)). For datasets without KC-Level data, we assume that KC and Question are the same, meaning one Question corresponds to one KC.

## 4.2 DATASETS AND BASELINE SETTINGS

We conduct experiments using the standard datasets processed by pyKT (Liu et al., 2022). For datasets that contain both exercises and the corresponding knowledge concepts, such as ASSISTments2009[1] and Algebra2005 [2], we evaluate the AUC and ACC metrics at both KC-level and Question-Level. For datasets that contain only questions or knowledge concepts, such as ASSISTments2015[3], we only evaluate the corresponding level. Table 1 summarizes the basic information of the datasets.We use 13 KT models for performance comparison, including DKT (Piech et al., 2015), DKT+ (Yeung & Yeung, 2018), DKT-F (Nagatani et al., 2019), KQN (Lee & Yeung, 2019), DKVMN (Zhang et al., 2017), ATKT (Guo et al., 2021), GKT (Nakagawa et al., 2019), SAKT (Pandey & Karypis, 2019), SAINT (Choi et al., 2020), AKT (Ghosh et al., 2020), LPKT (Shen et al., 2021), SimpleKT (Liu et al., 2023b), and AT-DKT (Liu et al., 2023a).

## 4.3 EXPERIMENTAL RESULT

### 4.3.1 COMPARATIVE ANALYSIS

We implemented the PyTorch model using the described method on the PYKT platform. After tuning the hyperparameters with wandb[4], we conducted experiments following the standard five-fold cross-validation procedure. The experimental results are shown in Tables 2 and 3, where Table 2 presents the AUC results at the Question Level and KC Level, and Table 3 presents the ACC results. The experimental results for DKT, DKT+, DKT-F, KQN, DKVMN, ATKT, GKT, SAKT, SAINT, and AKT are from the literature (Liu et al., 2022), while the results for LPKT and SIMPLEKT are from the literature (Liu et al., 2023b). The "-" indicates that the results were not reported in the original literature.

The data in Tables 2 and 3 show that compared to all baseline models, ExerCAKT achieved the best performance on the AS2009, AL2005, and AS2015 datasets. Specifically, in the Question Level evaluation, ExerCAKT achieved AUC scores of 0.7874 and 0.8312 on the AS2009 and AL2005 datasets, respectively, which are improvements of 4.41% and 2.00% compared to the DKT model. In the KC Level evaluation, ExerCAKT achieved excellent AUC scores of 0.7704 and 0.8249 on the AS2009 and AL2005 datasets, respectively, which are improvements of 0.70% and 1.95% compared to the AKT model. On the AS2015 datasets, which contain only knowledge concepts, the AUC score is 0.7287, respectively, outperforming most baseline models. Regarding the ACC metrics, the overall situation is similar to that of the AUC metrics, with ExerCAKT outperforming all baseline models on the AS2009, AL2005 and AS2015 datasets. The experimental results indicate that in both the Question Level and KC Level evaluations, all the datasets show that ExerCAKT achieved the best performance and outperformed the strong baseline model AKT, fully demonstrating the feasibility and excellent performance of ExerCAKT.

---

[1]https://sites.google.com/site/assistmentsdata/home/2009-2010-assistment-data/skill-builder-data-2009-2010

[2]https://pslcdatashop.web.cmu.edu/KDDCup/

[3]https://sites.google.com/site/assistmentsdata/datasets/2015-assistments-skill-builder-data

[4]https://wandb.ai/

Table 2: AUC index performance of ExerCAKT and other baseline models under 3 different datasets

| Model | KC Level(ALL-in-One) | | Question Level(ALL-in-One) | | AS2015 | ExerCAKT |
| | AS2009 | AL2005 | AS2009 | AL2005 | | #win/#tie/#loss |
|---|---|---|---|---|---|---|
| **DKT** | 0.7419 | 0.8146 | 0.7541 | 0.8149 | 0.7271 | 5/0/0 |
| **DKT+** | 0.7424 | 0.8144 | 0.7547 | 0.8156 | 0.7285 | 5/0/0 |
| **DKT-F** | - | 0.8163 | - | 0.8147 | - | 2/0/0 |
| **KQN** | 0.7361 | 0.8005 | 0.7477 | 0.8027 | 0.7254 | 5/0/0 |
| **DKVMN** | 0.7330 | 0.7891 | 0.7473 | 0.8054 | 0.7227 | 5/0/0 |
| **ATKT** | 0.7337 | 0.7964 | 0.7470 | 0.7995 | 0.7245 | 5/0/0 |
| **GKT** | 0.7227 | 0.8025 | 0.7424 | 0.8110 | 0.7258 | 5/0/0 |
| **SAKT** | 0.7085 | 0.7682 | 0.7246 | 0.7880 | 0.7114 | 5/0/0 |
| **SAINT** | 0.6865 | 0.6662 | 0.6958 | 0.7775 | 0.7026 | 5/0/0 |
| **AKT** | 0.7650 | 0.8091 | 0.7853 | 0.8306 | 0.7281 | 5/0/0 |
| **LPKT** | - | - | 0.7814 | 0.8274 | - | 2/0/0 |
| **AT-DKT** | - | - | - | 0.8246 | - | 1/0/0 |
| **SIMPLEKT** | - | - | 0.7744 | 0.8254 | 0.7248 | 3/0/0 |
| **ExerCAKT** | **0.7704** | **0.8249** | **0.7874** | **0.8312** | **0.7287** | - |
| **ΔDKT** | 0.0285 | 0.0103 | 0.0333 | 0.0163 | 0.0016 | - |

Table 3: ACC index performance of ExerCAKT and other baseline models under 3 different datasets

| Model | KC Level(ALL-in-One) | | Question Level(All-in-One) | | AS2015 | ExerCAKT |
| | AS2009 | AL2005 | AS2009 | AL2005 | | #win/#tie/#loss |
|---|---|---|---|---|---|---|
| **DKT** | 0.7181 | 0.7882 | 0.7244 | 0.8097 | 0.7503 | 5/0/0 |
| **DKT+** | 0.7191 | 0.7889 | 0.7248 | 0.8097 | 0.7510 | 5/0/0 |
| **DKT-F** | - | 0.7891 | - | 0.8090 | - | 5/0/0 |
| **KQN** | 0.7179 | 0.7850 | 0.7228 | 0.8025 | 0.7500 | 5/0/0 |
| **DKVMN** | 0.7144 | 0.7778 | 0.7199 | 0.8027 | 0.7508 | 5/0/0 |
| **ATKT** | 0.7158 | 0.7774 | 0.7208 | 0.7998 | 0.7494 | 5/0/0 |
| **GKT** | 0.7077 | 0.7825 | 0.7153 | 0.8088 | 0.7504 | 5/0/0 |
| **SAKT** | 0.7017 | 0.7729 | 0.7063 | 0.7954 | 0.7474 | 5/0/0 |
| **SAINT** | 0.6885 | 0.7538 | 0.6936 | 0.7791 | 0.7438 | 5/0/0 |
| **AKT** | 0.7323 | 0.7939 | 0.7392 | 0.8124 | 0.7521 | 5/0/0 |
| **LPKT** | - | - | 0.7355 | 0.8145 | - | 2/0/0 |
| **AT-DKT** | - | - | - | 0.8144 | - | 1/0/0 |
| **SIMPLEKT** | - | - | 0.7320 | 0.8083 | 0.7508 | 3/0/0 |
| **ExerCAKT** | **0.7350** | **0.7995** | **0.7405** | **0.8155** | **0.7525** | - |
| **ΔDKT** | 0.0169 | 0.0113 | 0.0161 | 0.0058 | 0.0022 | - |

### 4.3.2 ABLATION ANALYSIS

To verify the effectiveness of the context-aware + Rasch exercise representation method used in the ExerCAKT framework, we conducted ablation experiments in AS2009 dataset. The experimental results are shown in Table 4. Under the optimal hyperparameters, we set four experimental conditions and reported the results using five-fold cross-validation. The experimental conditions were set as follows:(I) Do not use context-aware and Rasch, i.e., use $e_{q_t}$ directly as the exercise representation. (II) Do not use context-aware, i.e., do not use GRU to learn the contextual representation of exercises, use $e_{q_t}$ for Rasch embedding. (III) Do not use Rasch embedding, i.e., use $d_{q_t}$ as the exercise representation. (IV) Use both context-aware and Rasch, i.e., the complete method described in this paper.

The experimental results indicate that (1) Condition IV outperforms all other conditions. When both context-aware and Rasch are used simultaneously, the model achieves optimal performance, demonstrating the effectiveness of this combination. (2) Condition I is superior to Condition II, suggesting that using Rasch embedding alone not only fails to improve performance but also leads to a performance decline. This might be due to the fact that Rasch embedding linearly adds the knowledge concept features and the exercise variation vector, and exercises that appear less frequently or involve multiple knowledge concepts may face difficulties in learning the embedding parameters,

Table 4: Ablation Study

| Model | Question Level(All-in-One) | | KC Level(ALL-in-One) | |
|---|---|---|---|---|
| | AUC | ACC | AUC | ACC |
| I.w/o Context-aware and Rasch | 0.7695 | 0.7289 | 0.7620 | 0.7246 |
| II.w/o Context-aware | 0.7586 | 0.7277 | 0.7454 | 0.7231 |
| III.ExerCAKT w/o Rasch | 0.7698 | 0.7299 | 0.7623 | 0.7259 |
| IV.ExerCAKT | 0.7874 | 0.7405 | 0.7704 | 0.7350 |

thereby affecting overall performance. (3) Condition III outperforms Condition I, indicating that the model shows a slight improvement when using context-aware alone. (4) Condition IV outperforms both Conditions II and III, suggesting that the combination of context-aware and Rasch is necessary for beneficial effects. The context-aware exercise sequence features should be considered as variations of exercises around the knowledge concepts. In Condition III, this sequence feature is directly used as the exercise feature, resulting in no performance improvement. The beneficial effect of the context-aware Rasch embedding is that, when encountering low-frequency or multi-concept exercises, the exercise variation vector comprehensively considers the entire sequence, allowing the model to still make effective predictions.

### 4.3.3 KEY HYPERPARAMETER ANALYSIS

The ExerCAKT framework introduces a hyperparameter $\gamma$ to control the proportion of specific vectors in exercise representation. To explore the impact of $\gamma$ on the model, we examined the performance changes in the $[0, 1]$ interval with a step size of 0.1 under different embedding dimensions and GRU layers settings. Due to the large number of hyperparameter combinations and limited computational hardware, we used one training set, validation set, and test set from the AS2009 dataset's five-fold cross-validation. The overall experimental visualization results are shown in Figure 3. (with ratio indicating the hyperparameter $\gamma$). This result shows the Question Level AUC performance under different hyperparameter settings. The way the hyperparameters $\gamma$ are set determines the way the lines are connected, and the color of the lines indicates the model's performance. The yellower the line, the higher the model's AUC, and the bluer the line, the lower the performance. The result shows that the lines are darker at $\gamma = 0.0, 0.1, 0.9, 1.0$, indicating poorer performance. It is important to note that a ratio of 0.0 means not considering the specificity of the exercises, while a ratio of 1.0 means using context-aware exercise sequence features as exercise embedding. To further analyze the impact of this hyperparameter $\gamma$ on the model's performance, we plotted a box plot of all the experimental data, as shown in Figure 4. The horizontal axis represents different parameter settings, and the vertical axis represents the achieved performance. The maximum, median, and minimum performance values are labeled in red, black, and blue fonts, respectively.

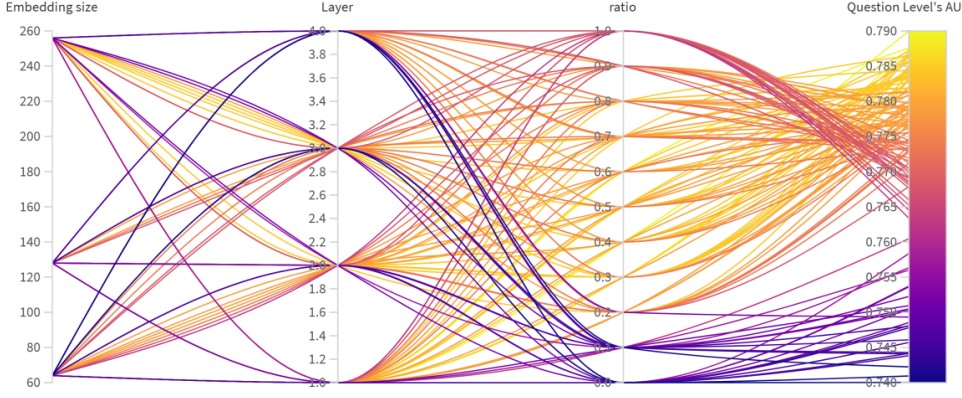

Figure 3: The model's Question Level AUC performance under different embedding dimensions, GRU layers, and ratio parameters.

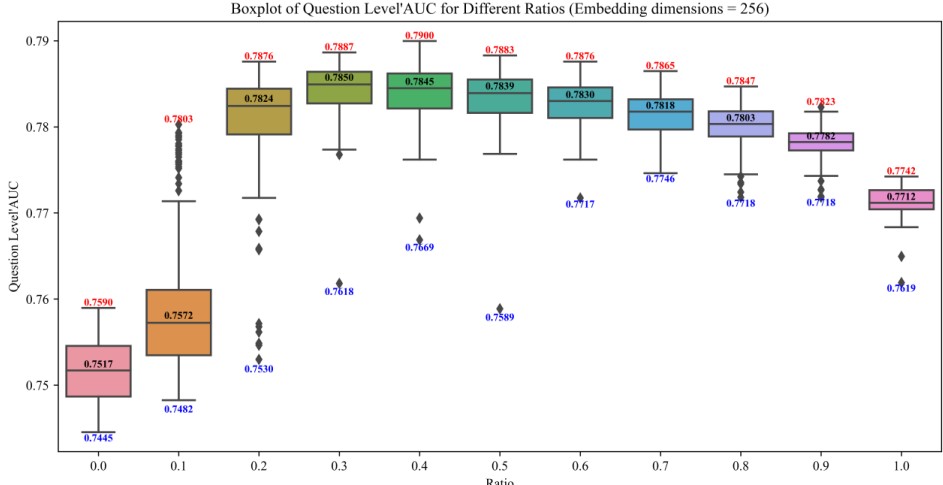

Figure 4: Box plot of the model's Question Level AUC performance under different ratio parameters.

The results in Figure 4 show that as the $\gamma$ increases, the model's performance first improves and then declines. In all experimental results, the difference between the maximum and minimum values of the Question Level AUC exceeds 4%, indicating that this parameter significantly affects the model's performance. Within the $[0.3, 0.6]$ interval, the AUC achieves a maximum value of 0.7900 with a median exceeding 0.7830, indicating that the model performs best in this parameter range with fewer outliers, demonstrating stable performance. For ratios of $\gamma = 0.0, 0.1, 0.9, 1.0$, the median is below 0.7800, and the model's performance is poorer. Specifically, when $\gamma = 0.1$ or $\gamma = 0.2$, the model has more outliers and a larger range of anomalies, indicating that with a smaller proportion of specific vectors, the model's performance becomes unstable. The model's performance in KC Level AUC metrics, which is similar to the Question Level AUC metrics but with more outliers in the $[0.6, 0.9]$ interval. Based on the analysis, the model performs better and is more stable in the $[0.3, 0.5]$ interval.

## 5 CONCLUSION

This paper presents the ExerCAKT model, which effectively models knowledge states and provides high-quality representations of exercises through GRU-based knowledge state and exercise feature extractors. The paper demonstrates the competitive performance of ExerCAKT through comparative experiments on three datasets, two evaluation methods, and two evaluation metrics. Additionally, ablation and hyperparameter experiments validate the effectiveness of the module design and the framework's scalability. Future research could focus on optimizing knowledge state modeling and exercise representation in ExerCAKT, such as: (1) proposing new RNN structures to replace GRU units; (2) optimizing exercise representation by considering characteristics such as exercise difficulty and multi-knowledge-concept associations based on historical data; (3) incorporating additional optimization objectives to improve model accuracy and generalization.

## REPRODUCIBILITY STATEMENT

Our code is available in the URL (now it is anonymous, and we submitted the anonymous code in the supplementary materials during review). All experiments are conducted through pyKT, and baseline models and data sets can be obtained through it. The code provided by us includes the configuration of ablation experiments, and provides a super parameter search function to maximize the replication of the experiments involved in this paper.

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

## A   APPENDIX

The appendix includes details of the dataset and baseline models and further experiments. In addition to the contrast experiment, ablation experiment and super parameter experiment in the text, we also further discussed three issues.

Q1: The ExerCAKT framework uses two GRUs as the Knowledge State Feature Extractor and Exercise Feature Extractor. Would replacing GRUs with RNNs or LSTMs yield similar performance?

Q2: What is the impact of other hyperparameters on model performance?

Q3: What are the inference speed and computational overhead of ExerCAKT?

### A.1   DATASET INFORMATION

#### A.1.1   ASSISTMENTS2009

The ASSISTments2009 dataset (hereafter referred to as AS2009) is used for student modeling and knowledge tracing research. It collects data from exercises on the online tutoring platform ASSISTments during the 2009-2010 academic year. The dataset includes student interaction records during problem-solving, sequence data, exercise data, and the knowledge concepts (KCs) involved. It provides real-world scenario data for evaluating student modeling and knowledge tracing algorithms, and is widely used to assess and compare the performance and effectiveness of various knowledge tracing methods. In the processed dataset, there are on average 1.1970 knowledge concepts associated with each exercise, totaling 337,415 interactions, 4,661 sequences, 17,737 exercises, and 123 different knowledge concepts.

#### A.1.2   ALGEBRA2005

The Algebra2005 dataset (hereafter referred to as AL2005) is a dataset from the KDD Cup 2010 EDM Challenge, containing responses from 13-14 year-old students to algebra exercises. The processed Algebra2005 dataset includes 884,098 interactions, involving 4,712 sequences, 173,113 exercises, and 112 knowledge concepts (KCs), with an average of 1.3634 knowledge concepts associated with each exercise.

#### A.1.3   ASSISTMENTS2015

The ASSISTments2015 dataset is similar to the ASSISTments2009 dataset, as both come from the online tutoring platform ASSISTments. The difference is that the ASSISTments2015 dataset is derived from platform records from the year 2015. The processed dataset includes 682,789 interactions, involving 19,292 sequences, and does not include specific exercise information, only 100 knowledge concepts.

### A.2   BASELINE MODELS INFORMATION

#### A.2.1   DKT (PIECH ET AL., 2015)

The DKT model is a model that predicts students' knowledge states using Long Short-Term Memory (LSTM) networks. It inputs students' historical response sequences into the LSTM to capture changes and patterns in students' knowledge states. These knowledge states can be used to predict students' performance on new exercises.

### A.2.2 DKT+ (YEUNG & YEUNG, 2018)

The DKT+ model is an improvement upon the DKT model, aiming to address two issues in deep knowledge tracing. The model introduces regularization to improve the accuracy and stability of predictions regarding students' performance. The DKT+ model shows better performance and robustness in knowledge tracing tasks, providing more precise personalized teaching support in the educational field.

### A.2.3 DKT-F (NAGATANI ET AL., 2019)

The DKT-F model improves upon the DKT model by considering that knowledge acquired early by learners may be forgotten as interactions continue to occur.

### A.2.4 KQN (LEE & YEUNG, 2019)

The KQN model explores the interaction between knowledge and skills. It introduces a Knowledge Query Network to better capture the relationship between students' knowledge mastery and skill development.

### A.2.5 DKVMN (ZHANG ET AL., 2017)

The DKVMN model is a Dynamic Key-Value Memory Network. It uses memory storage to track students' knowledge mastery. By inputting problems and students' response sequences, the model can dynamically retrieve and update the knowledge information stored in memory. The model predicts students' responses to new exercises based on previous answers, the current problem, and the stored knowledge state.

### A.2.6 ATKT (GUO ET AL., 2021)

The ATKT model enhances knowledge tracing through adversarial training. It uses adversarial learning principles to improve the KT model's generalization ability by jointly training on the original inputs and corresponding adversarial examples.

### A.2.7 GKT (NAKAGAWA ET AL., 2019)

The GKT model is a knowledge tracing model based on Graph Neural Networks. It represents students' knowledge states and the relevance of exercises using graph structures. Through message passing and node updating in the graph, the GKT model dynamically captures changes in students' knowledge and predicts future learning outcomes.

### A.2.8 SAKT (PANDEY & KARYPIS, 2019)

The SAKT model is a knowledge tracing model based on self-attention mechanisms. It uses a self-attention mechanism that automatically focuses on key information in students' response sequences to accurately predict students' performance on future exercises.

### A.2.9 SAINT (CHOI ET AL., 2020)

The SAINT model predicts students' knowledge levels by calculating queries, keys, and values. It uses an Encoder-Decoder structure to accurately capture key information for each exercise when processing students' response sequences, thereby improving the prediction of students' future performance.

### A.2.10 AKT (GHOSH ET AL., 2020)

AKT (Context-Aware Attentive Knowledge Tracing) is a context-aware attention mechanism knowledge tracing model. It uses Rasch models and interaction-distance-based attention mechanisms to model learners' current states, better capturing individual differences and changes in knowledge.

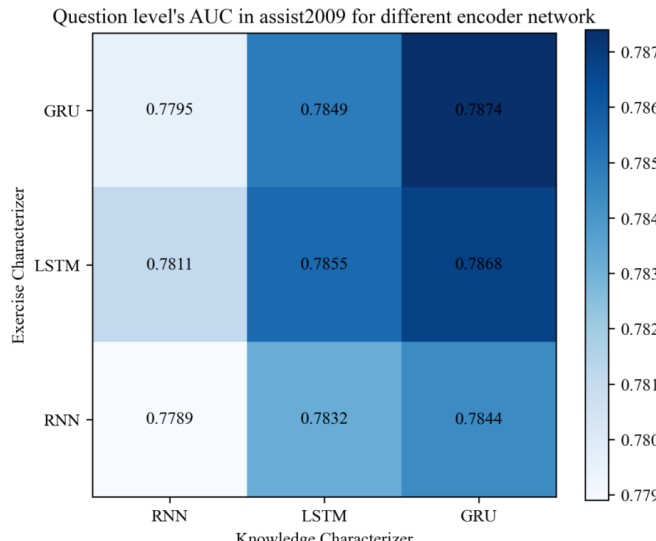

Figure 5: The impact of different feature encoders on the Question Level AUC performance on the AS2009 dataset.

### A.2.11 LPKT (SHEN ET AL., 2021)

The LPKT model simulates the student learning process. It replicates learning and forgetting processes similar to those in LSTM and GRU, and uses a Q-matrix to describe the relationship between exercises and knowledge concepts.

### A.2.12 SIMPLEKT (LIU ET AL., 2023B)

The SIMPLEKT model is a simple yet effective knowledge tracing baseline model. It uses improved Rasch embeddings and ordinary dot-product attention to achieve knowledge tracing, providing good scalability.

### A.2.13 AT-DKT (LIU ET AL., 2023A)

AT-DKT enhances model prediction performance by introducing auxiliary tasks related to exercise labeling and personalized prior knowledge prediction. It proposes two auxiliary tasks, demonstrating improved model performance and showing that incorporating multi-task objectives is effective in the knowledge tracing field.

### A.3 RQ1

Q1:The ExerCAKT framework uses two GRUs as the Knowledge State Feature Extractor and Exercise Feature Extractor. Would replacing GRUs with RNNs or LSTMs yield similar performance?

ExerCAKT can easily replace GRU units with RNN or LSTM units. We explored the impact of different recurrent neural units on the model by replacing the GRU units in both feature extractors, to uncover potential avenues for future improvements.

Figure 5 and Figure 6 display heatmaps showing the results of Question Level AUC and KC Level AUC under different combinations of RNN, LSTM, and GRU. The results indicate that: (1) The difference between the maximum and minimum values for both Question Level AUC and KC Level AUC is 0.85% and 0.71%, respectively. This difference suggests that there may be significant variations depending on the combination used, with the maximum values observed in the [GRU, GRU] combination, confirming the effectiveness of using dual GRU units in ExerCAKT. (2) The minimum values for both metrics are 0.7789 and 0.7633, which are higher than most baseline models, demonstrating that various recurrent neural network units perform well under this framework. (3) When

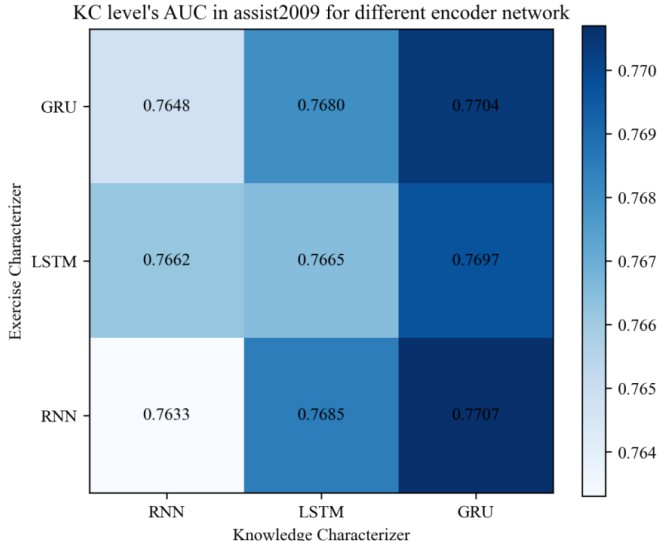

Figure 6: The impact of different feature encoders on the KC Level AUC performance on the AS2009 dataset.

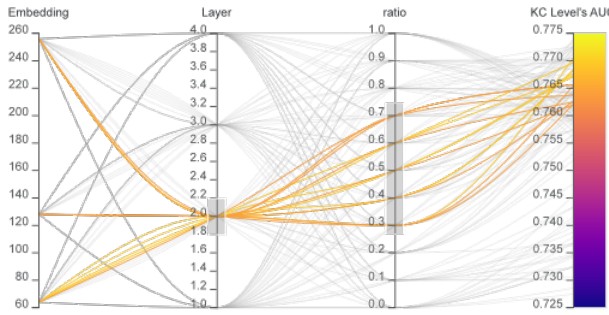

Figure 7: Schematic of the 15 experimental results when the number of GRU layers is 2.

the knowledge state feature encoder uses GRU, the model performs better compared to the other two configurations. In contrast, when the exercise feature extractor uses GRU, the model performance varies more significantly, and tends to increase with the choice of knowledge state feature encoder. This indicates that the choice of the knowledge state feature extractor is more critical, although the choice of exercise feature extractor can also improve model performance. Overall, we found that different recurrent neural network units yield different results. In fact, LSTM is an enhanced version of RNN, and GRU is a simplified version of LSTM. Given the variety of network structures, we believe that GRU might not be the optimal choice. Similar to the work of LPKT (Shen et al., 2021), designing more efficient and interpretable units could further enhance the model, with ExerCAKT providing a foundational framework.

## A.4 RQ2

Q2:What is the impact of other hyperparameters on model performance?

In addition to the selection of hyperparameters and feature extractors, we believe that embedding dimensions and the number of GRU layers might be important hyperparameters affecting model performance. Therefore, we conducted multiple experiments under different embedding dimensions, GRU layer counts, and hyperparameters $\gamma$. To exclude the interference from other hyperparameter settings, we selected experimental data within the hyperparameter range $\gamma \in [0.3, 0.7]$ , and computed the average performance for all GRU layer counts when exploring embedding dimensions.

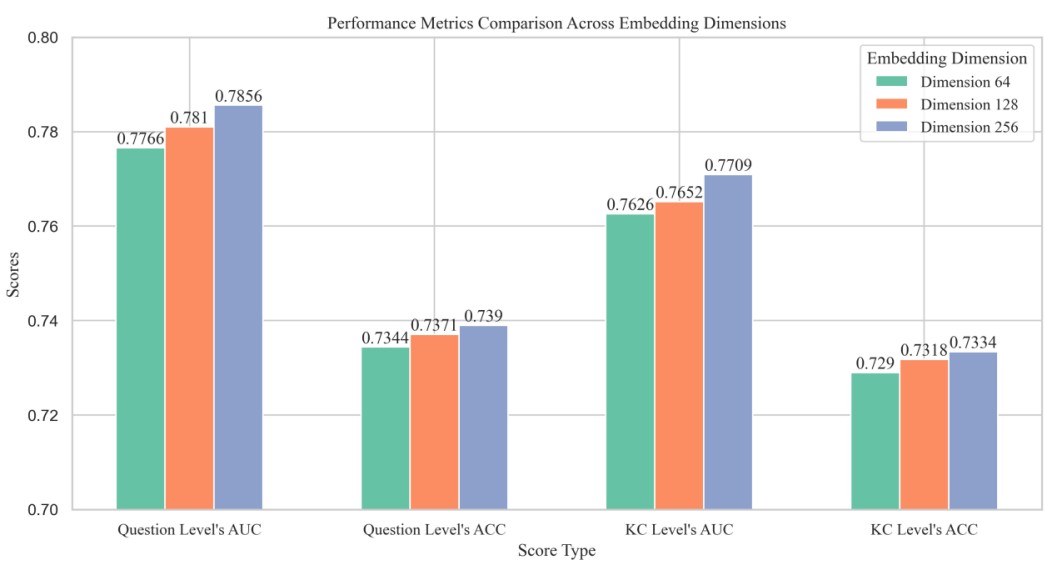

Figure 8: The model's performance under different embedding dimensions.

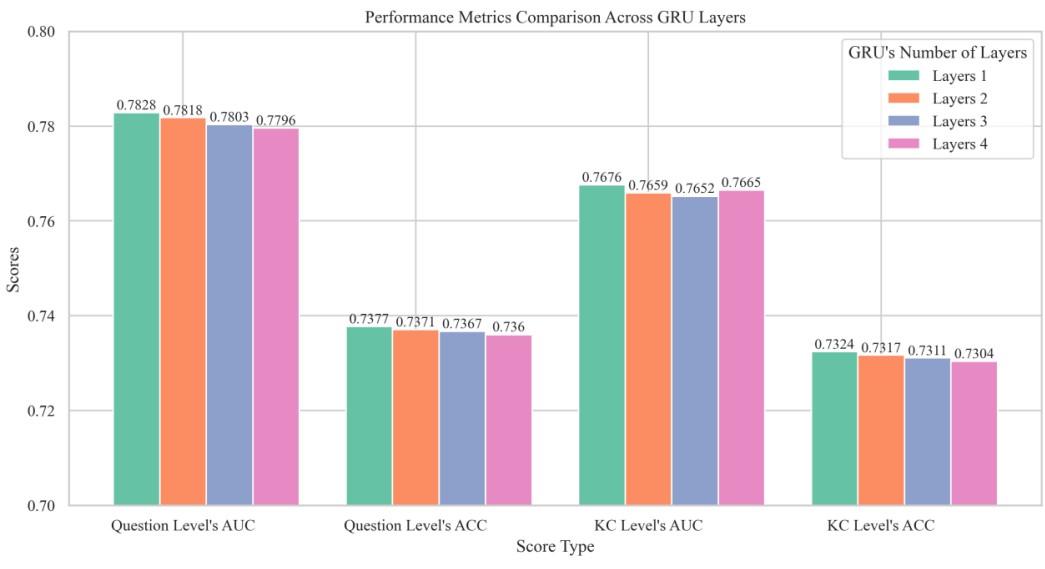

Figure 9: The model's performance under different GRU layer counts.

For example, when investigating the impact of different GRU layer counts on performance, we have three embedding dimensions: $[64, 128, 256]$, and four GRU layer counts: $[1, 2, 3, 4]$. Therefore, the performance for each GRU layer count is averaged over 15 results. Figure 7 shows the 15 experimental results when the number of GRU layers is 2, calculates their average, and reports the results.

Figure 8 and Figure 9 display the performance of AUC and ACC metrics for Question Level and KC Level under different embedding dimensions and GRU layer counts, respectively. It can be observed that embedding dimensions have a significant impact on model performance, with the AUC metric improving by 0.0090 as the embedding dimension increases. Regarding the number of GRU layers, the model achieves the best performance in multiple metrics with just one layer, indicating that the model does not require excessive stacking of layers.

Table 5: cost

| Model | Inference time cost (ms) | GPU memory usage (MB) |
|---|---|---|
| **LPKT** | 171.46 ms | 3367 MB |
| **SIMPLEKT** | **19.55 ms** | 2139 MB |
| **AKT** | 81.87 ms | 9567 MB |
| **ExerCAKT** | 27.99 ms | **1981 MB** |

## A.5 RQ3

Q3:What are the inference speed and computational overhead of ExerCAKT?

To evaluate model inference speed and GPU memory usage, we compared ExerCAKT, AKT, SIM-PLEKT, and LPKT on the Assistment2009 dataset. The hyperparameters were set to the optimal results obtained from the respective model searches, but the embedding dimension was fixed at 128 to ensure a fair comparison of GPU memory usage. Experiments were conducted individually on a workstation equipped with an I9-12900k CPU, RTX 3090 (24G) GPU, and 64G RAM.

The experimental results indicate that ExerCAKT has lower inference time costs compared to AKT, LPKT, and other models, though it is slightly inferior to SimpleKT. However, the difference is minimal. In terms of GPU memory usage, ExerCAKT outperforms the three comparison models. Although the datasets commonly used in knowledge tracing are still relatively small and the demand for high-concurrency knowledge tracing model inference has not yet emerged, the time cost during training is not as perceptible. However, GPU memory usage determines whether the algorithm can run on some consumer-grade GPUs (e.g., Nvidia RTX 3060 Ti 8G). As envisioned, future intelligent education algorithms may be widely deployed on edge computing nodes held by teachers and schools, which will protect data privacy. Consequently, the speed and hardware costs of algorithms will receive increased attention.

