# OpenReview forum: "ExerCAKT: A Knowledge Tracing Model Based on GRU Capturing Contextual Features of Exercises"
_ICLR.cc/2025/Conference — Submitted to ICLR 2025_

### Official Review · Reviewer_ffdZ · 2024-10-30

**Soundness:** 2
**Presentation:** 3
**Contribution:** 2
**Rating:** 5
**Confidence:** 3

**Summary:**

The authors focus on student knowledge state modeling and propose a GRU-based knowledge tracing model, ExerCAKT. The ExerCAKT model uses GRU to process students' knowledge states and exercise features (via a knowledge state feature extractor and an exercise feature extractor) and achieves exercise feature representation by contextual relationships + Rasch embedding. The authors evaluate ExerCAKT on three datasets—ASSISTments2009, Algebra2005, and ASSISTments2015—and the experimental results demonstrate that ExerCAKT outperforms baseline models, including Transformer-based and RNN-based models, in both AUC and ACC metrics at the KC and question levels. Additionally, the authors conduct an ablation experiment to verify the effectiveness of contextual relationships + Rasch embedding.

Contributions:
1. Proposing a GRU-based knowledge tracing model, ExerCAKT.
2. Introducing a method that combines contextual relationships and Rasch embedding. ExerCAKT generates exercise representations by integrating changes in students' historical interaction sequences with Rasch embedding, which characterizes exercise difficulty. This integration enables the model to achieve more accurate predictions.
3. Validating the model's stability, accuracy, and robustness through comparative and ablation experiments.

**Strengths:**

Originality:
The paper proposes a GRU-based knowledge tracking model, ExerCAKT, which enhances the representation of knowledge states and exercise features by combining contextual relationships and Rasch embedding. This approach captures relationships and changes across different knowledge concepts, thereby improving prediction accuracy.

Quality:
The comparison with various baseline models on the ASSISTments2009, Algebra2005, and ASSISTments2015 datasets demonstrates the effectiveness of ExerCAKT. The ablation experiment  evaluates the independent effects of contextual relationships + Rasch embedding and verifies the advantages of their combined application.

Clarity:
The paper presents a clear structure and logical flow, detailing each step from problem definition to model framework and experimental validation. The authors provide explanations for choosing the GRU model, as well as for using contextual relationships + Rasch embedding.

Significance:
This research is significant in the field of knowledge tracking. The ExerCAKT model performs well on multiple datasets, surpassing some existing baseline models, and illustrates the potential of a GRU-based deep learning framework for knowledge tracking.

**Weaknesses:**

Insufficient ablation experiments on the model framework:
Although the authors verify the effectiveness of combining contextual relationships with Rasch embedding, the ablation studies on the main components of the overall model framework—specifically, the 'exercise feature extractor' and 'knowledge state feature extractor'—are not fully explored. The current framework relies on GRU as the core of these two components without further investigation into whether alternative architectures (such as LSTMs, Transformers, CNNs, etc.) could provide performance improvements for the same tasks. Additionally, the authors do not compare models using different architectures (such as GRU, LSTMs, Transformers, CNNs, etc.) with other baseline models and do not fully evaluate the effectiveness of these components when supported by different architectures.

Insufficient explanation for the combination of Rasch embedding and contextual relationships:
Although the authors demonstrate that combining contextual relationships with Rasch embeddings can improve model performance, they do not fully explain why this combination is necessary. The rationale for combining contextual relationships with Rasch embeddings is not fully explained, and providing a more  theoretical justification along with specific examples of their interaction would improve the clarity of the paper. Additionally, testing this combination on a single dataset is insufficient to validate its effectiveness. The authors could extend this analysis by conducting experiments on different datasets and examining the individual and combined effects of contextual relationships and Rasch embeddings for a more comprehensive evaluation.

Insufficient breadth of dataset selection:
Although the authors demonstrate the effectiveness of ExerCAKT on the AS2009, AL2005, and AS2015 datasets, there are some limitations in the dataset selection. The criteria for dataset selection are unclear, some baseline models lack results on the chosen datasets, and the authors do not explain how they handled this incomplete comparison. Besides the three datasets used, the PYKT platform supports additional datasets (e.g., Statics2011, Bridge2006, Ednet, NIPS34, POJ, Statics2011), which the authors could consider including some of them to enable a more comprehensive and balanced comparison.

**Questions:**

Question 1: The current ablation experiments examine only the combination of contextual relationships and Rasch embedding. Have the authors considered testing alternative architectures (such as LSTMs, Transformers, or CNNs) as the core components of the exercise feature extractor and knowledge state feature extractor? Evaluating the model’s performance with these different architectures could provide a more comprehensive view of potential improvements.

Question 2: This paper uses GRU to capture student interaction information but does not compare it with other sequence modeling architectures (like LSTMs or Transformers) across multiple datasets. Are there plans to assess these different architectures on additional datasets, and could the authors discuss where GRU may offer advantages over other approaches in capturing interaction patterns?

Question 3: Could the authors clarify their criteria for selecting the AS2009, AL2005, and AS2015 datasets for evaluation and explain how they address the issue of missing comparison results for certain baseline models on these datasets?

Question 4: Given that some baseline models lack results on the current dataset selection, could the authors consider including additional datasets, such as Statics2011, Bridge2006, or Ednet, to ensure that all compared models have complete experimental results?

---

### Official Review · Reviewer_rmhz · 2024-11-03

**Soundness:** 3
**Presentation:** 3
**Contribution:** 2
**Rating:** 3
**Confidence:** 4

**Summary:**

The authors present a "simple" model based on GRU that outperforms existing deep learning approaches for knowledge tracing.

**Strengths:**

The ablation study was clear and precise. I enjoyed Figure 3.

**Weaknesses:**

I do not agree that DKT is a "extremely simple model" given the number of parameters it has. Wilson et al. (2016) managed to match the performance of DKT with retrained IRT, which is clearly a very simple model.

> Wilson, Kevin H., et al. "Back to the basics: Bayesian extensions of IRT outperform neural networks for proficiency estimation." in Educational Data Mining 2016: 539.

If I compare to Gervet et al. 2020, I see that your approach underestimates SAKT and that a logistic regression (Best LR) matches the top performance of ExerCAKT for AL2015 (question level) and outperforms ExerCAKT for AS2009 (KC level).

> Gervet, Theophile, et al. "When is deep learning the best approach to knowledge tracing?." Journal of Educational Data Mining 12.3 (2020): 31-54.

According to this other paper, the performance of IKT on Assistments 2009 is 0.797 and its performance on Algebra 2005 is 0.851. According to this same paper the performance of AKT on Algebra 2005 is 0.845 which would already outperform ExecCAKT.

> Minn, Sein, et al. "Interpretable knowledge tracing: Simple and efficient student modeling with causal relations." Proceedings of the AAAI conference on artificial intelligence. Vol. 36. No. 11. 2022.

I also encourage the authors to read about logistic knowledge tracing.

> Pavlik Jr, Philip Irvin, and Luke G. Eglington. "Automated search improves logistic knowledge tracing, surpassing deep learning in accuracy and explainability." Journal of Educational Data Mining 15.3 (2023): 58-86.

Minor comments:
- "accuracy.To" ane "objectives.The" in the abstract (missing spaces), same in the text "conditions.We", "comparison.In"
- One brace missing in Eq 10
- The typography of (1) (2) (3) in Section 3.3 is not pretty. There are no spaces.
- $1e-3$ is not pretty. $1\mathrm{e}{-3}$ would be better.

**Questions:**

1) What strucks me is that the results in Table 2 are identical than the results of simpleKT (Liu et al. 2023b) and the results of the NeurIPS 2022 pyKT paper (Liu et al. 2022). Does this mean that the results were fully deterministic and reproducible with the exact same random seed, or that the results were copypasted without any attempt to reproduce? According to the code, the users are shuffled with random seed 1024. Then I am wondering whether the results still hold with a different seed.

2) Why do you limit your baselines to deep learning models while there is a large body of research (see Weaknesses) that shows that logistic-based approaches are simpler, more interpretable, and sometimes more accurate than their deep learning counterparts?

---

### Official Review · Reviewer_JcKn · 2024-11-04

**Soundness:** 2
**Presentation:** 2
**Contribution:** 1
**Rating:** 3
**Confidence:** 4

**Summary:**

This paper suggests a simple model for the knowledge tracing (KT) task, which utilizes gated recurrent units (GRUs) to model knowledge states and exercise features to extract more effective exercise representations for prediction. Experimental results on three small-scale datasets showcase that the proposed ExerCAKT holds slightly better performance than more than ten compared KT models.

**Strengths:**

- Clear motivation for this work: inventing a simple yet effective model for KT.

- Convincing experiments regarding the number of compared baseline KT models. Benefiting from the pyKT platform, the authors compared more than ten KT models with the proposed model in their experiments.

- The source code of this work is available.

**Weaknesses:**

- Insufficient review of related work. The review of deep learning-based KT models is too simplistic, somewhat outdated, and not convincing. Moreover, there have been several similar works to the proposed ExerCAKT, but the authors did not report or discuss them. For instance, the approach [1] is very similar to the proposed approach: it is also a simple KT model based on a forget-response-update mechanism, which models students’ states at three levels. As shown in  [1], it seems to be more comprehensive than those of the proposed approach.
```
[1] Shen, X., Yu, F., Liu, Y., Liang, R., Wan, Q., Yang, K., & Sun, J. (2024, October). Revisiting Knowledge Tracing: A Simple and Powerful Model. In Proceedings of the 32nd ACM International Conference on Multimedia (pp. 263-272).
```
- The discussion about more Transformer-based or Transformer-like KT approaches (not limited to the approaches contained in pyKT) is missing, such as DTransformer [2], SAINT+ [3], and HiTSKT [4].
```
[2] Yin, Yu, et al. “Tracing Knowledge Instead of Patterns: Stable Knowledge Tracing with Diagnostic Transformer.” Proceedings of the ACM Web Conference. 2023.
[3] Shin, D., Shim, Y., Yu, H., Lee, S., Kim, B., & Choi, Y. (2021, April). Saint+: Integrating temporal features for ednet correctness prediction. In LAK21: 11th International Learning Analytics and Knowledge Conference (pp. 490-496).
[4] Ke, Fucai, et al. "HiTSKT: A hierarchical transformer model for session-aware knowledge tracing." Knowledge-Based Systems 284 (2024): 111300.
```
- Lack of more convincing datasets for KT. In general, EdNet  and SLP  were originally collected for KT, which are much more challenging than some common datasets due to their scale and data variety. On these datasets, Transformer-based KT models, such as the representative SAINT, have shown promising performance compared to common KT models. However, there are no corresponding results to report such comparisons on these challenging datasets. Therefore, the initial assumptions and conclusions drawn from the current experimental results may not be convincing.

- The authors argue that complex models are not necessary. Although some experimental results were reported to validate this assumption, no theoretical analysis is provided to demonstrate it.

- In my opinion, this work is not an innovative effort to tackle significant issues or create novel models based on theoretical analysis. It is much more like a tutorial presenting comparisons between existing models and a tuned model with optimal hyperparameters through a well-built open-source platform. Furthermore, no interesting phenomena can be drawn from the results. Therefore, the contributions and novelty of this work seem insufficient.

- There are some writing typos, grammar errors, and inconsistent terms in this manuscript. Additionally, the statistical test methods utilized in Tables 2 and 3 are not specified.

**Questions:**

See above.

---

### Official Review · Reviewer_eGeX · 2024-11-06

**Soundness:** 2
**Presentation:** 2
**Contribution:** 1
**Rating:** 3
**Confidence:** 4

**Summary:**

This paper introduces a knowledge tracing model based on GRU designed to capture contextual features of exercises effectively. It aims to predict student performance through historical interactions, leveraging the strengths of simple RNNs in handling sequential data. The experiments on three public datasets show the effectiveness of the method.

**Strengths:**

1. The paper presents a novel GRU-based knowledge tracing model that demonstrates strong performance compared to existing baselines.
2. The authors have conducted comprehensive experiments and provided a detailed analysis of their model’s performance.

**Weaknesses:**

1. The paper lacks novelty and omits crucial baselines. Transformer-based knowledge tracing models have already been extensively studied, as seen in  DTransformer[1]. DTransformer builds the architecture from the question level to the knowledge level, which closely aligns with the modeling approach in this paper. The authors should refer to their methods and provide relevant discussion and explanation.

2. One of the key points highlighted in the paper is that overly complex models may not be necessary to achieve excellent performance in knowledge tracing tasks. However, the conclusion drawn from results on only three datasets is not convincing to me. It is important to validate the findings on a broader range of datasets, such as Statics2011, Junyi, and EdNet-KT4. Additionally, metrics beyond AUC and ACC, like MAE and RMSE, should be used for further validation.


[1]Yu Yin, Le Dai, Zhenya Huang, Shuanghong Shen, Fei Wang, Qi Liu, Enhong Chen, Xin Li, Tracking Knowledge Instead of Patterns: Stable Knowledge Tracing with Diagnostic Transformer, The Web Conference 2023 (WWW'2023): 855-864, Austin, Texas, USA, April 30-May 4, 2023

**Questions:**

1.  Could the authors elaborate on how ExerCAKT might be adapted to handle more complex educational datasets or different types of knowledge tracing tasks?
2.  How does the scalability of ExerCAKT compare to other models, especially when dealing with larger datasets or more complex knowledge structures?

---

### Meta-Review · Area_Chair_xVdz · 2024-12-21

**Metareview:**

This work proposes a GRU-based knowledge tracing model, ExerCAKT, which utilizes GRU to process students' knowledge states and exercise features. Some experiments demonstrate its effectiveness. However, the authors fail to provide a thorough literature review of related works, and the motivation for the study lacks novelty. Additionally, the experiments are insufficient in terms of baselines, datasets, and evaluation metrics. Overall, I recommend that the authors carefully revise and improve this work based on the reviewers' comments.

**Additional Comments On Reviewer Discussion:**

The author didn't respond to Reviewers.

---

### Decision · Program_Chairs · 2025-01-22

Reject